# Towards Efficient Contrastive PAC Learning

**Jie Shen**                                                                                 *jieshen.sjtu@gmail.com*
*Department of Computer Science*
*Stevens Institute of Technology*

**Reviewed on OpenReview:** *https://openreview.net/forum?id=dBJo9hyKVg*

## Abstract

We study contrastive learning under the PAC learning framework. While a series of recent works have shown statistical results for learning under contrastive loss, based either on the VC-dimension or Rademacher complexity, their algorithms are inherently inefficient or not implying PAC guarantees. In this paper, we consider contrastive learning of the fundamental concept of linear representations. Surprisingly, even under such basic setting, the existence of efficient PAC learners is largely open. We first show that the problem of contrastive PAC learning of linear representations is intractable to solve in general. We then show that it can be relaxed to a semi-definite program when the distance between contrastive samples is measured by the $\ell_2$-norm. We then establish generalization guarantees based on Rademacher complexity, and connect it to PAC guarantees under certain contrastive large-margin conditions. To the best of our knowledge, this is the first efficient PAC learning algorithm for contrastive learning.

## 1 Introduction

Contrastive learning has been a successful learning paradigm in modern machine learning (Gutmann & Hyvärinen, 2010; Logeswaran & Lee, 2018). In general, it is assumed that a learner has access to an anchor example $x$, a positive example $y$, and a number of negative examples $\{z_1, \ldots, z_k\}$, and the goal of contrastive learning is to learn a representation function $f$ on the examples such that $y$ is closer to $x$ than all $z_i$'s under $f$.

Motivated by the empirical success of contrastive learning, there have been a surge of recent interests that attempt to understand it from a theoretical perspective, primarily through the lens of Rademacher complexity or that of VC-theory. For example, Arora et al. (2019) initiated the study of generalization ability of contrastive learning by analyzing the Rademacher complexity of a commonly used contrastive loss, and showed that under certain structural assumptions on the data, minimizing the unsupervised contrastive loss leads to a low classification error. There were a few follow-up works in this line which aimed to understand and improve the sample complexity; see e.g. (Ash et al., 2022; Awasthi et al., 2022; Lei et al., 2023).

Orthogonal to the Rademacher-based theory, a very recent work of Alon et al. (2024) proposed to study this problem under the classical probably approximately correct (PAC) learning framework (Valiant, 1984). Unlike prior works that assumed a rich structure for the data distribution in order to estimate the classification error from contrastive loss, Alon et al. (2024) considered that there is an unknown distribution on the instances and labels, where labels are produced by an unknown distance function. Tight bounds on sample complexity were established for arbitrary distance functions, $\ell_p$-distances, and tree metrics.

In this work, we follow the contrastive PAC learning framework of Alon et al. (2024). Let $\mathcal{X} \subset \mathbb{R}^d$ be the space of examples (i.e. image patches). An instance $u$ is a tuple $(x, y, z) \in \mathcal{X}^3$; thus we denote by $\mathcal{U} := \mathcal{X}^3$. The label, $b$, of a tuple $(x, y, z)$ is either $-1$ or $1$; here, we write $\mathcal{B} := \{-1, 1\}$ as the label space. Let $\mathcal{H} := \{h : \mathcal{U} \to \mathcal{B}\}$ be a hypothesis class. Suppose that there is an unknown distribution $D$ on $\mathcal{U} \times \mathcal{B}$. We are mainly interested in the realizable setting in this paper, namely, there exists an $h^* \in \mathcal{H}$, such that for all $(u, b) \sim D$, it holds almost surely that $b = h^*(u)$. Now for any hypothesis $h \in \mathcal{H}$, we can define its error

rate as follows: $\mathrm{err}_D(h) := \mathrm{Pr}_{(u,b)\sim D}(h(u) \neq b) = \mathrm{Pr}_{u\sim D_U}(h(u) \neq h^*(u))$, where $D_U$ denotes the marginal distribution of $D$ on $\mathcal{U}$. We are now in the position to define the contrastive PAC learning problem.

**Definition 1** (Contrastive PAC learning). Let $\epsilon, \delta \in (0,1)$ be a target error rate and failure probability, respectively. An adversary $\mathrm{EX}(D, h^*)$ chooses a distribution $D_U$ on $\mathcal{U}$ and $h^* \in \mathcal{H}$ and fixes them throughout the learning process. Each time the learner requests a sample from the adversary, the adversary draws a sample $u$ from $D_U$, labels it by $b := h^*(u)$ and returns $(u, b)$ to the learner. The goal of the learner is to find a concept $\hat{h} : \mathcal{U} \to \mathcal{B}$, such that with probability at least $1 - \delta$ (over the random draws of samples and all internal randomness of the learning algorithm), it holds that $\mathrm{err}_D(\hat{h}) \leq \epsilon$ for all $D, h^*$.

One example of the hypothesis class is $\mathcal{H} = \{h : (x, y, z) \mapsto \mathrm{sign}\big(\left\|f(x) - f(z)\right\|_p - \left\|f(x) - f(y)\right\|_p\big)\}$, where both $f(\cdot)$ and $p$ are to be learned from samples. This is a contrastive PAC learning problem considered in Alon et al. (2024). We note that since learning distance functions is inherently challenging, the PAC guarantees of Alon et al. (2024) were established only for finite domains, i.e. $|\mathcal{X}|$ is finite, and the learning algorithm is inherently inefficient. On the other side, Arora et al. (2019) and many of its follow-up works such as Awasthi et al. (2022); Lei et al. (2023) considered a fixed and known distance function, e.g. $p = 2$, and aimed to learn the representation function $f(\cdot)$ among a certain family. This makes the problem more tractable, though in general, it is still inefficient due to the non-convexity of the contrastive loss – only convergence to stationary points is known (Yuan et al., 2022). In addition, the approaches in this line were not immediately implying PAC guarantees.

In this paper, we investigate the contrastive PAC learning problem for fixed $p = 2$ and we aim to develop computationally efficient algorithms with PAC guarantees. Our setup is thus interpolating Arora et al. (2019) and Alon et al. (2024). Despite the relatively new setup, it is surprising that even efficient contrastive PAC learning for linear representation functions on $\mathbb{R}^d$ is largely open. Indeed, as to be shown later, this is already a non-trivial problem from the computational perspective.

From now on, we will focus on the very fundamental class of linear representation functions:

$$\mathcal{F} = \{f_W : x \mapsto Wx, W \in \mathcal{W}\}. \tag{1.1}$$

In the above, $\mathcal{W}$ can be certain constraint set such as the Frobenius ball. We will discuss in more detail the choice of $\mathcal{W}$ and related results later. Denote

$$g_W(x, y, z) := \left\|Wx - Wz\right\|_2^2 - \left\|Wx - Wy\right\|_2^2. \tag{1.2}$$

Now we can spell out the hypothesis class to be learned:

$$\mathcal{H} = \big\{h_W : (x, y, z) \mapsto \mathrm{sign}\big(g_W(x, y, z)\big), W \in \mathcal{W}\big\}. \tag{1.3}$$

## 1.1 Main results

Our main results for contrastive PAC learning of (1.3) is as follows.

**Theorem 2** (Theorem 10, informal). *Suppose that $b \cdot g_{W^*}(x, y, z) \geq 1$ for all $(x, y, z, b) \sim D$. There exists an algorithm $\mathcal{A}$ satisfying the following. By drawing $\mathrm{poly}(1/\epsilon, \log 1/\delta)$ samples from $D$, with probability $1 - \delta$, $\mathcal{A}$ outputs a hypothesis $\hat{W}$ such that $\mathrm{err}_D(\hat{W}) \leq \epsilon$. In addition, $\mathcal{A}$ runs in $\mathrm{poly}(1/\epsilon, \log 1/\delta)$ time.*

We remark that the condition $b \cdot g_{W^*}(x, y, z) \geq 1$ is similar to the large-margin condition for learning halfspaces. Such large-margin condition was broadly assumed to analyze performance of learning algorithms such as Perceptron (Rosenblatt, 1958) and boosting (Schapire & Freund, 2012). Our condition is adapted to the contrastive samples, and we will call it contrastive large-margin condition. The constant 1 therein can be readily replaced by a margin parameter $\gamma > 0$, which will then lead to a sample complexity proportional to $1/\gamma^2$ by our analysis. This is standard in learning theory (Anthony & Bartlett, 1999). However, to keep our results concise, we do not pursue it here.

Our sample complexity in Theorem 2 omits dependence on other quantities such as the magnitude of samples and the size of the constraint set $\mathcal{W}$. A complete description can be found in Theorem 10.

What we really hope to highlight in the informal version is that we developed a polynomial-time algorithm that PAC learns a fundamental concept class from contrastive samples, and this is the first efficient PAC learner in the literature.

## 1.2 Overview of our techniques

We first view the contrastive PAC learning problem as binary classification, as suggested in (1.3). We then apply standard learning principles such as empirical risk minimization with a suitable loss function. It turns out, however, that the quadratic form of $g_W$ makes the problem inherently intractable even under the hinge loss function. We thus make use of the property that quadratic functions can be linearized by introducing a new matrix variable, which turns the problem into a semi-definite program (SDP) that can be solved in polynomial time. In order to analyze the error rate, we establish generalization bounds via Rademacher complexity on the SDP. We then show that with the contrastive large-margin condition, the empirical risk goes to zero on the target concept $W^*$. This implies that the error rate of a solution of the SDP can be as small as $\epsilon$. Lastly, we apply eigenvalue decomposition on the SDP solution to obtain a linear representation, which completes the proof.

## 1.3 Roadmap

A concrete problem setup as well as a collection of useful notations are presented in Section 2. In Section 3, we elaborate on our algorithm and the theoretical guarantees. Section 4 concludes this paper and proposes a few open questions.

## 2 Preliminaries

The PAC learning framework was proposed by Valiant (1984). Let $\mathcal{U}$ and $\mathcal{B}$ be the instance and label space, respectively. It is assumed that there is an underlying distribution $D$ on $\mathcal{U} \times \mathcal{B}$ such that all samples are drawn from $D$. Let $\mathcal{H}$ be a hypothesis class that maps $\mathcal{U}$ to $\mathcal{B}$. The error rate of $h \in \mathcal{H}$ is defined as $\mathrm{err}_D(h) := \Pr_{(u,b) \sim D}(h(u) \neq b)$. Under the realizable setting, there exists a target hypothesis $h^* \in \mathcal{H}$, such that with probability 1, $b = h^*(u)$ for $(u, b) \sim D$.

In contrastive learning, an instance $u \in \mathcal{U}$ is often a tuple of the form $u = (x, y, z)$, where $x, y, z$ are from $\mathcal{X} \subset \mathbb{R}^d$. For example, $\mathcal{X}$ can be the space of image patches with size $d$, and $u$ consists of three image patches. More generally, $u$ may contain a number of patches $x, y, z_1, \ldots, z_k$, where the $z_i$'s are often referred to as negative examples in the literature and $y$ is referred to as positive example. In our main results, we did not pursue such generalization to keep our algorithm and theory concise. However, it is known that such extension is possible and we will illustrate it in Section 3.

With $\mathcal{U} = \mathcal{X}^3$ and $\mathcal{B} = \{-1, 1\}$ in mind, a sample $(x, y, z, b)$ of contrastive learning should be interpreted as follows: if $b = 1$, it indicates that $y$ is closer to $x$ than $z$ is to; otherwise, $z$ is closer to $x$. More formally, there exists a distance function $\rho^* : \mathcal{X} \times \mathcal{X} \to \mathbb{R}_{\geq 0}$, such that $h^*(x, y, z) = \mathrm{sign}(\rho^*(x, z) - \rho^*(x, y))$. We note that Alon et al. (2024) aimed to learn such general distance functions over a finite domain, while most prior works assumed certain parameterized form such as $\rho^*(x, y) = \|Wx - Wy\|_2^2$, as in this work. Once we confine ourselves to the specific distance function, we can think of the mapping $Wx$ as a new representation of $x$. Thus, sometimes the problem of contrastive learning is also regarded as representation learning. Denote

$$g_W(x, y, z) = \|Wx - Wz\|_2^2 - \|Wx - Wy\|_2^2. \tag{2.1}$$

Observe that $h^*(x, y, z) = \mathrm{sign}(g_{W^*}(x, y, z))$.

As typical in machine learning, one may want to impose certain constraint on $W$ in order to prevent overfitting. Of particular interest would be the Frobenius-norm ball $\mathcal{W}_F = \{W \in \mathbb{R}^{d' \times d} : \|W\|_F \leq r_F\}$, the $\ell_1$-norm ball $\mathcal{W}_1 = \{W \in \mathbb{R}^{d' \times d} : \|W\|_1 \leq r_1\}$ for sparsity, or the nuclear-norm ball $\mathcal{W}_* = \{W \in \mathbb{R}^{d' \times d} : \|W\|_* \leq r_*\}$ for low-rankness. Different constraints will lead to different generalization bounds, which will be shown in Section 3.

For a square matrix $M$, we write $\mathrm{tr}(M)$ for its trace. The inner product of two matrices $A$ and $B$ with same size is defined as $\langle A, B \rangle := \mathrm{tr}(A^\top B)$, where sometimes we simply write as $A \cdot B$. In addition to the matrix norms that are just mentioned, we may also use the spectral norm; it is denoted by $\|M\|$.

We will mainly be interested in the hinge loss as a surrogate function to the error rate. Denote

$$L(W; U) = \max\{0, 1 - W^\top W \cdot U\}, \qquad \tilde{L}(G; U) = \max\{0, 1 - G \cdot U\}. \tag{2.2}$$

The $W$, $G$, and $U$ will be matrices in this paper.

Let $\mathcal{F}$ be a class of real-valued functions on $\mathcal{U} \times \mathcal{B}$ and $S = \{s_i\}_{i=1}^n$ be a sample set of $\mathcal{U} \times \mathcal{B}$. The empirical Rademacher complexity of $S$ under $\mathcal{F}$ is defined as $\mathcal{R}(\mathcal{F} \circ S) := \frac{1}{n}\mathbb{E}_\sigma \sup_{f \in \mathcal{F}} \sigma_i \cdot f(s_i)$, where $\sigma = (\sigma_1, \ldots, \sigma_n)$ is the Rademacher random vector.

## 3 Algorithms and Performance Guarantees

Let $S = \{(x_i, y_i, z_i, b_i)\}_{i=1}^n$ be a set of samples independently drawn from $D$, where the tuple $(x_i, y_i, z_i) \in \mathcal{U}$ and $b_i \in \mathcal{B}$. Recall that we study the realizable PAC learning. Thus there exists an unknown $W^* \in \mathcal{H}$ such that for all $(x, y, z, b) \sim D$, $b = g_{W^*}(x, y, z)$.

At first glance, one may seek a hypothesis $W \in \mathcal{H}$ that minimizes the empirical risk. That is,

$$\min_{W \in \mathcal{W}} \frac{1}{n} \sum_{i=1}^n \mathbf{1}[b_i \cdot g_W(x_i, y_i, z_i) < 0], \tag{3.1}$$

where $\mathbf{1}[E]$ is the indicator function which outputs 1 if the event $E$ occurs and 0 otherwise.

Since $g_W(\cdot)$ is quadratic in $W$, it is easy to show by algebraic calculation that:

**Lemma 3.** $\|Wx - Wy\|_2^2 = \langle W^\top W, (x - y)(x - y)^\top \rangle$.

Therefore, let

$$U_i = b_i \cdot ((x_i - z_i)(x_i - z_i)^\top - (x_i - y_i)(x_i - y_i)^\top). \tag{3.2}$$

We can obtain

$$b_i \cdot g_W(x_i, y_i, z_i) = \langle W^\top W, U_i \rangle.$$

Plugging the above into (3.1) gives

$$\min_{W \in \mathcal{W}} \frac{1}{n} \sum_{i=1}^n \mathbf{1}[\langle W^\top W, U_i \rangle < 0]. \tag{3.3}$$

Unfortunately, solving the above program is intractable, due to the 1) non-convexity of the 0/1-loss, and 2) the quadratic formula with respect to $W$. In the following, we propose approaches based on semi-definite programming, that is solvable in polynomial time.

First, by standard technique, we could alternatively minimize the hinge loss:

$$\min_{W \in \mathcal{W}} \frac{1}{n} \sum_{i=1}^n L(W; U_i), \tag{3.4}$$

where $L(\cdot; \cdot)$ was defined in (2.2). We note that Verma & Branson (2015) also studied the above loss function in the context of Mahalanobis distance metrics, and they obtained statistical sample complexity. Observe that the problem may still be non-convex, since $U_i$ may have negative eigenvalues – this is in stark contrast to learning from standard examples. Since the non-convexity comes from the quadratic term $W^\top W$, we consider replacing the variable $W^\top W$ with a new variable $G$. Hence, $\langle G, U_i \rangle$ is a linear function with respect to $G$, turning the objective function into convex. This is a well-known technique that has been used in many contexts (Williamson & Shmoys, 2011; d'Aspremont et al., 2007).

Suppose that based on the fact $W \in \mathcal{W}$, we obtain a constraint set $\mathcal{G} \supset \{W^\top W : W \in \mathcal{W}\}$. As far as $\mathcal{G}$ is constructed as convex, the overall program becomes convex. Note that such convex set $\mathcal{G}$ always exists, and the minimal is called convex hull (Boyd & Vandenberghe, 2004). The empirical risk minimization program that we are going to analyze is given as follows:

$$\min_{G \in \mathcal{G}} \frac{1}{n} \sum_{i=1}^{n} \tilde{L}(G; U_i), \tag{3.5}$$

where $\tilde{L}(\cdot; \cdot)$ was defined in (2.2).

## 3.1 Rademacher complexity

We provide bounds on the Rademacher complexity of (3.5) for two popular choices of $\mathcal{W}$ (and thus $\mathcal{G}$).

### 3.1.1 Frobenius-norm ball

We first consider the Frobenius-norm ball, one of the most widely used constraints in machine learning. That is, $\mathcal{W} = \mathcal{W}_F := \{W \in \mathbb{R}^{d' \times d} : \|W\|_F \leq r_F\}$ for some parameter $r_F > 0$. Here and after, the subscript of $\mathcal{W}$ and $r$ is used only to identify the type of constraints. Since $G = W^\top W$, by singular value decomposition, it is not hard to show that $\|G\|_* \leq r_F^2$ where $\|\cdot\|_*$ denotes the nuclear norm (also known as the trace norm). Therefore, we can choose

$$\mathcal{G} = \mathcal{G}_* := \{G \in \mathbb{R}^{d \times d} : G \succeq 0, \|G\|_* \leq r_F^2\}. \tag{3.6}$$

**Lemma 4.** *Consider the function class $\Theta_* := \{\theta : U \mapsto \tilde{L}(G; U), G \in \mathcal{G}_*\}$. Let $S = \{U_i\}_{i=1}^n$ and assume $\max_{U_i \in S} \|U_i\| \leq \alpha$. Then the empirical Rademacher complexity*

$$\mathcal{R}(\Theta_* \circ S) \leq r_F^2 \cdot \alpha \cdot \sqrt{\frac{\log d}{n}}.$$

*Proof.* Let $\sigma = (\sigma_1, \ldots, \sigma_n)$ be the Rademacher random variable. By the contraction property of Rademacher complexity, we have

$$n \cdot \mathcal{R}(\Theta_* \circ S) = \mathbb{E}_\sigma \sup_{G \in \mathcal{G}_*} \sum_{i=1}^{n} \sigma_i \max\{0, 1 - G \cdot U_i\}$$

$$\leq \mathbb{E}_\sigma \sup_{G \in \mathcal{G}_*} \sum_{i=1}^{n} \sigma_i G \cdot U_i$$

$$\leq r_F^2 \cdot \max_{U_i \in S} \|U_i\| \cdot \sqrt{n \log d},$$

where the last inequality follows from Kakade et al. (2012) (see Table 1 therein). The result follows by noting that the spectral norm of $U_i$ is assumed to be upper bounded by $\alpha$. $\qquad\square$

Recall that $U_i$ was defined in (3.2). Suppose that the example space $\mathcal{X}$ is a subset of a bounded $\ell_2$-norm ball, say $\mathcal{X} \subset \{x : \|x\|_2 \leq \kappa\}$. Then we can show that

$$\|U_i\| \leq \|x_i - y_i\|_2^2 + \|x_i - z_i\|_2^2 \leq 2\kappa^2.$$

Thus setting $\alpha = 2\kappa^2$ in Lemma 4 gives the following:

**Corollary 5.** *Consider the function class $\Theta_* := \{\theta : U \mapsto \tilde{L}(G; U), G \in \mathcal{G}_*\}$. Suppose $\mathcal{X} \subset \{x : \|x\|_2 \leq \kappa\}$ and let $S = \{U_i\}_{i=1}^n$ be a draw of sample set from $\mathcal{X}^3$. Then*

$$\mathcal{R}(\Theta_* \circ S) \leq 2r_F^2 \cdot \kappa^2 \cdot \sqrt{\frac{\log d}{n}}.$$

### 3.1.2 $\ell_1$-norm ball (sparsity)

Now we consider that the linear representation matrix $W$ is constrained by an $\ell_1$-norm, which typically promotes sparsity patterns (Tibshirani, 1996; Chen et al., 1998; Candès & Tao, 2005). That is, $\mathcal{W} = \mathcal{W}_F :=$ $\{W \in \mathbb{R}^{d' \times d} : \|W\|_1 \leq r_1\}$ for some parameter $r_1 > 0$. Now we derive the $\ell_1$-norm for $W^\top W$. To do so, let us write $W$ in a column form: $W = (w_1, \dots, w_d)$ where $w_i$ denotes the $i$-th column of $W$. It follows that

$$
\begin{aligned}
\left\| W^\top W \right\|_1 &= \sum_{1 \leq i,j \leq d} |w_i \cdot w_j| \\
&\leq \sum_{1 \leq i,j \leq d} \|w_i\|_1 \cdot \|w_j\|_\infty \\
&= \sum_{1 \leq j \leq d} \|w_j\|_\infty \sum_{1 \leq i \leq d} \|w_i\|_1 \\
&\leq \sum_{1 \leq j \leq d} \|w_j\|_1 \cdot r_1 \leq r_1^2.
\end{aligned}
$$

This suggests that we could choose

$$
\mathcal{G} = \mathcal{G}_1 := \{G \in \mathbb{R}^{d \times d} : G \succeq 0, \|G\|_1 \leq r_1^2\}. \tag{3.7}
$$

**Lemma 6.** *Consider the function class $\Theta_1 := \{\theta : U \mapsto \tilde{L}(G; U), G \in \mathcal{G}_1\}$. Let $S = \{U_i\}_{i=1}^n$ and assume $\max_{U_i \in S} \|U_i\|_\infty \leq \alpha$. Then the empirical Rademacher complexity*

$$
\mathcal{R}(\Theta_1 \circ S) \leq r_1^2 \cdot \alpha \cdot \sqrt{\frac{4 \log(2d)}{n}}.
$$

*Proof.* For a matrix $M$, let $\vec{M}$ be the vector obtained by concatenating all columns of $M$.

Let $\sigma = (\sigma_1, \dots, \sigma_n)$ be the Rademacher random variable. By the contraction property of Rademacher complexity, we have

$$
\begin{aligned}
n \cdot \mathcal{R}(\Theta_1 \circ S) &= \mathbb{E}_\sigma \sup_{G \in \mathcal{G}_1} \sum_{i=1}^n \sigma_i \max\{0, 1 - G \cdot U_i\} \\
&\leq \mathbb{E}_\sigma \sup_{G \in \mathcal{G}_1} \sum_{i=1}^n \sigma_i G \cdot U_i \\
&= \mathbb{E}_\sigma \sup_{G \in \mathcal{G}_1} \sum_{i=1}^n \sigma_i \vec{G} \cdot \vec{U}_i \\
&\leq r_1^2 \cdot \alpha \cdot \sqrt{2n \log(2d^2)},
\end{aligned}
$$

where the last inequality follows from Lemma 26.11 of Shalev-Shwartz & Ben-David (2014). Dividing both sides by $n$ completes the proof. □

Suppose that $\mathcal{X} \subset \{x : \|x\|_\infty \leq \kappa\}$. Then we can show that

$$
\|U_i\|_\infty \leq \|x_i - y_i\|_\infty^2 + \|x_i - z_i\|_\infty^2 \leq 2\kappa^2.
$$

Therefore, specifying $\alpha = 2\kappa^2$ in the above lemma gives the following corollary.

**Corollary 7.** *Consider the function class $\Theta_1 := \{\theta : U \mapsto \tilde{L}(G; U), G \in \mathcal{G}_1\}$. Suppose $\mathcal{X} \subset \{x : \|x\|_\infty \leq \kappa\}$ and let $S = \{U_i\}_{i=1}^n$ be a draw of sample set from $\mathcal{X}^3$. Then*

$$
\mathcal{R}(\Theta_1 \circ S) \leq 2r_1^2 \cdot \kappa^2 \cdot \sqrt{\frac{4 \log(2d)}{n}}.
$$

## 3.2 PAC guarantees

We analyze the PAC guarantees under a new type of margin condition, which we call the contrastive large-margin condition.

**Definition 8** (Contrastive large-margin condition)**.** We say that the data distribution $D$ satisfies the contrastive large-margin condition if there exists $W^* \in \mathcal{W}$, such that for all $(x, y, z, b) \sim D$, the following holds with probability 1: $b(\|W^*x - W^*z\|_2^2 - \|W^*x - W^*y\|_2^2) \geq 1$.

Geometrically, this condition ensures that there is a non-trivial separation between positive examples and negative examples for any given anchor $x$. It follows that when the condition is satisfied, (3.4) attains an optimal objective value of 0. Since the feasible set of convex program of (3.5) contains that of (3.4), it is easy to get the following.

**Lemma 9.** *Assume that the contrastive large-margin condition holds. Then there exists $\hat{G} \in \mathcal{G}$, such that the objective value of* (3.5) *at $\hat{G}$ equals 0.*

Now we can prove the main result of this section, the PAC guarantees.

**Theorem 10.** *Assume that the contrastive large-margin condition is satisfied for some $W^* \in \mathcal{W}$, and $\mathcal{G}$ is such that $\mathcal{G} \supset \{W^\top W : W \in \mathcal{W}\}$. Let $\hat{G} \in \mathcal{G}$ be an optimal solution to* (3.5) *and let $\hat{G} = V\Sigma V^\top$ be its eigenvalue decomposition. Let $\hat{W} := \Sigma^{1/2}V^\top$. Then by drawing contrastive sample set $S = \{(x_i, y_i, z_i, b_i)\}_{i=1}^n$, with probability at least $1 - \delta$, it holds that*

$$\mathrm{err}_D(\hat{W}) \leq 2\mathcal{R}(\Theta \circ S) + 5c\sqrt{\frac{2\log(8/\delta)}{n}},$$

*where $c := \sup_{G \in \mathcal{G}, U \in \mathcal{U}} \left|\tilde{L}(G; U)\right|$. When $\mathcal{G}$ is a convex set, our algorithm runs in polynomial time.*

*Proof.* Let $\Theta := \{\theta : U \mapsto \tilde{L}(G; U), G \in \mathcal{G}\}$. Let $c := \sup_{G \in \mathcal{G}, U \in \mathcal{U} \times \mathcal{B}} \left|\tilde{L}(G; U)\right|$.

We apply standard uniform concentration via Rademacher complexity (Bartlett & Mendelson, 2002) to obtain that with probability $1 - \delta$,

$$\mathbb{E}_{U \sim D}\tilde{L}(\hat{G}; U) \leq \mathbb{E}_{U \sim D}L(W^*; U) + 2\mathcal{R}(\Theta \circ S) + 5c\sqrt{\frac{2\log(8/\delta)}{n}}.$$

In view of the contrastive large-margin condition, we have $L(W^*; U) = 0$. On the other hand, we always have

$$\tilde{L}(G; U) \geq \mathbf{1}[\hat{G} \cdot U < 0].$$

This implies

$$\mathbb{E}_{U \sim D}\mathbf{1}[\hat{G} \cdot U < 0] \leq 2\mathcal{R}(\Theta \circ S) + 5c\sqrt{\frac{2\log(8/\delta)}{n}}. \tag{3.8}$$

Now recall that $U = b\big((x - z)(x - z)^\top - (x - y)(x - y)^\top\big)$ as in (3.2), and $\hat{G} = \hat{W}^\top \hat{W}$ by the eigenvalue decomposition. Therefore,

$$\hat{G} \cdot U = b \cdot \left(\left\|\hat{W}x - \hat{W}z\right\|_2^2 - \left\|\hat{W}x - \hat{W}y\right\|_2^2\right).$$

Thus, (3.8) is equivalent to

$$\mathrm{err}_D(\hat{W}) \leq 2\mathcal{R}(\Theta \circ S) + 5c\sqrt{\frac{2\log(8/\delta)}{n}}. \tag{3.9}$$

The proof is complete. $\qquad\qquad\square$

Theorem 10, in allusion to the Rademacher complexity bounds in Section 3.1, lead to the sample complexity bounds for efficient contrastive PAC learning.

**Corollary 11.** *Assume same conditions as in Theorem 10. Consider $\Theta = \Theta_*$ as in Corollary 5. Suppose $\mathcal{X} \subset \{x : \|x\|_2 \le \kappa\}$. Then by drawing $n = \left(\frac{5+5r_F^2\kappa^2}{\epsilon}\right)^2 \log \frac{8d}{\delta}$ contrastive samples from $D$, we have $\mathrm{err}_D(\hat{W}) \le \epsilon$ with probability $1 - \delta$, where $\hat{W}$ is defined in Theorem 10.*

*Proof.* We just need to compute the supremum of $\left|\tilde{L}(G; U)\right|$. It turns out that $\left|\tilde{L}(G; U)\right| \le 1 + |G \cdot U| \le 1 + \|G\|_* \cdot \|U\| \le 1 + r_F^2\kappa^2$. The result follows by plugging this upper bound and the Rademacher complexity in Corollary 5 into Theorem 10. $\qquad\square$

**Corollary 12.** *Assume same conditions as in Theorem 10. Consider $\Theta = \Theta_1$ as in Corollary 7. Suppose $\mathcal{X} \subset \{x : \|x\|_\infty \le \kappa\}$. Then by drawing $n = \left(\frac{5+5r_1^2\kappa^2}{\epsilon}\right)^2 \log \frac{8d}{\delta}$ contrastive samples from $D$, we have $\mathrm{err}_D(\hat{W}) \le \epsilon$ with probability $1 - \delta$, where $\hat{W}$ is defined in Theorem 10.*

*Proof.* Again, we only need to compute the supremum of $\left|\tilde{L}(G; U)\right|$. It turns out that $\left|\tilde{L}(G; U)\right| \le 1 + |G \cdot U| \le 1 + \|G\|_1 \cdot \|U\|_\infty \le 1 + r_1^2\kappa^2$. The result follows by plugging this upper bound and the Rademacher complexity in Corollary 7 into Theorem 10. $\qquad\square$

We remark that both $\Theta_*$ and $\Theta_1$ are convex sets, thus our PAC guarantees are obtained from a computationally efficient algorithm, i.e. the convex program (3.5).

### 3.3 Extension to multiple negative examples

One important extension of our contrastive PAC learning framework is to consider multiple negative samples, which are commonly used in practice and its importance has been broadly studied (Ash et al., 2022; Awasthi et al., 2022; Lei et al., 2023). That is, suppose the label $b = 1$, in addition to the anchor example $x$ and a positive example $y$, a learner collects $k$ negative examples $z_1, \ldots, z_k$. Together, these serve as a sample $u := (x, y, z_1, \ldots, z_k, 1)$. Therefore, the instance space $U = \mathcal{X}^{k+2}$ while the label space remains same as before. The learning paradigm still follows from Definition 1. More generally, one can think of an instance as $(x, u_1, \ldots, u_{k+1})$ and a label $b \in \{1, \ldots, k+1\}$ that specifies the index among all $u_i$'s that is closest to $x$. Since we can always reorder the examples $u_1, \ldots, u_{k+1}$ such that the closest example is arranged at the first place, without loss of generality, we will always assume $b = 1$ and the example following $x$ is closest, which we denote as $y$, and the remaining examples are denoted by $z_1, \ldots, z_k$. This is also a notation typically seen in the literature.

Now given a set of contrastive samples $S = \{(x_i, y_i, z_{i1}, \ldots, z_{ik}, b_i)\}_{i=1}^n$ where the samples are independently drawn from $D$, we aim to establish PAC guarantees as the case $k = 1$. For any $i$, we know by the realizability assumption that $\|W^*x_i - W^*y_i\|_2 \le \|W^*x_i - W^*z_{ij}\|_2$ for all $1 \le j \le k$. Define

$$U_{ij} = (x_i - z_{ij})(x_i - z_{ij})^\top - (x_i - y_i)(x_i - y_i)^\top. \tag{3.10}$$

By Lemma 3, we have $\langle (W^*)^\top W^*, U_{ij} \rangle \ge 0$ for all $1 \le j \le k$. This is equivalent to $\min_{1 \le j \le k}\langle (W^*)^\top W^*, U_{ij} \rangle \ge 0$. Thus, a natural empirical risk, based on hinge loss, is as follows:

$$\min_{W \in \mathcal{W}} \frac{1}{n} \sum_{i=1}^n \max\{0, 1 - \min_{1 \le j \le k} \langle W^\top W, U_{ij} \rangle\}. \tag{3.11}$$

As discussed in the preceding subsection, the above program is non-convex, and we will consider SDP as convex relaxation. This gives the following program:

$$\min_{G \in \mathcal{G}} \frac{1}{n} \sum_{i=1}^n \max\{0, 1 - \min_{1 \le j \le k} \langle G, U_{ij} \rangle\}. \tag{3.12}$$

Consider the function class $\mathcal{Q} = \{q_G : (x, y, z_1, \ldots, z_k) \mapsto \max\{0, 1 - \min_{1 \le j \le k} G \cdot U_{\cdot j}\}, G \in \mathcal{G}\}$, where $U_{\cdot j} = (x - z_j)(x - z_j)^\top - (x - y)(x - y)^\top$. Let $c := \sup_{G \in \mathcal{G}, (x,y,z_1,\ldots,z_k) \in \mathcal{X}^{k+2}} |q_G(x, y, z_1, \ldots, z_k)|$ and denote

$\hat{G}$ a global optimum of (3.12). Write $u = (x, y, z_1, \ldots, z_k)$. Then standard concentration results tell that

$$\mathbb{E}_{u \sim D} q_{\hat{G}}(u) \leq \mathbb{E}_{u \sim D} q_{G^*}(u) + 2\mathcal{R}(\mathcal{Q} \circ S) + 5c\sqrt{\frac{2\log(8/\delta)}{n}},$$

where $G^* = (W^*)^\top W^*$. Under the contrastive large-margin condition, we have $q_{G^*}(u) = 0$. Thus, it remains to bound the empirical Rademacher complexity $\mathcal{R}(\mathcal{Q} \circ S)$. To this end, we think of the function $q_G \in \mathcal{Q}$ as a composition of two functions: $q_G = \tilde{q} \circ \bar{q}_G$, where $\bar{q}_G(u) = (G \cdot U_{\cdot 1}, \ldots, G \cdot U_{\cdot k}) \in \mathbb{R}^k$ and $\tilde{q}(v_1, \ldots, v_k) = \max\{0, 1 - \min_{1 \leq j \leq k} v_j\}$. By Corollary 4 of Maurer (2016), we have

$$n \cdot \mathcal{R}(\mathcal{Q} \circ S) \leq \sqrt{2} L \mathbb{E}_\sigma \sup_{G \in \mathcal{G}} \sum_{i=1}^{n} \sum_{j=1}^{k} \sigma_{ij} G \cdot U_{ij}, \tag{3.13}$$

where $L$ denotes the Lipschitz constant of $\tilde{q}$.

When $\mathcal{G} = \mathcal{G}_*$ and $\mathcal{X} \subset \{x : \|x\|_2 \leq \kappa\}$, we have shown that the expectation on right-hand side is less than $\sqrt{nk \log d} \cdot r_F^2 \kappa^2$. Therefore, it remains to estimate $L$. Observe that $\tilde{q}$ can further be thought of as $\tilde{q}(t) = \max\{0, 1 - t\}$ and $t = \min_{1 \leq j \leq k} v_j$. The Lipschitz constant of $t$ with respect to $(v_1, \ldots, v_k)$ is upper bounded by 1. Thus, $L = 1$.

Putting together gives

$$\mathbb{E}_{u \sim D} q_{\hat{G}}(u) \leq 2\sqrt{2} r_F^2 \kappa^2 \sqrt{\frac{k \log d}{n}} + 5c\sqrt{\frac{2\log(8/\delta)}{n}} \tag{3.14}$$

when $\mathcal{G} = \mathcal{G}_*$. We note that $c = 1 + r_F^2 \kappa^2$ by algebraic calculation.

Lastly, similar to the proof of Theorem 10, the above implies PAC guarantee of $\hat{W}$ with $\hat{G} = \hat{W}^\top \hat{W}$.

## 4 Conclusion and Open Questions

In this paper, we studied the power of convex relaxations for contrastive PAC learning. We showed that even for learning linear representations via contrastive learning, the problem is generally intractable, which is in stark contrast to the classic problem of PAC learning linear models. We then proposed a convex program based on techniques from semi-definite programming. Under a contrastive large-margin condition, we proved that the solution to the convex program enjoys PAC guarantees.

This is the first work that establishes PAC guarantees for contrastive learning for arbitrary domain, while the very recent work is confined to finite domains (and considers a more involved learning scenario). Our convex relaxation techniques seem suitable for the $\ell_2$-distance between contrastive samples. An important question is whether there exists more general approach to dealing with other distance metrics such as the $\ell_1$-distance. We expect that this is possible, since $\ell_1$-norm is closely related to a family of linear functions by introducing additional variables. Another important question is whether it is possible to learn nonlinear representation functions, for example, the family of polynomial threshold functions or neural networks. We conjecture that learning neural networks from contrastive samples is rather challenging, since the optimization landscape for linear classes is already drastically changed with contrastive samples. On the algorithmic design front, it appears that one needs to carefully design convex surrogate functions whenever the underlying representation functions are modified. Does there exist a principled approach that guides the design, and is it necessary to consider convex surrogate functions for the problem? In the literature of PAC learning halfspaces, there have been a rich set of algorithmic results showing that one may optimize certain non-convex loss functions whose stationary point really enjoys PAC guarantees (Diakonikolas et al., 2020; Zhang et al., 2020; Shen, 2021a; 2025). Can we show similar results for contrastive PAC learning? In particular, can we design non-convex loss functions that may serve as a proxy to (3.1) and that a good stationary point can be efficiently found? We believe that our work will serve as a first step towards these questions. Lastly, it is known that in practice, the contrastive examples and labels can both be noisy. Is it possible to develop noise-tolerant algorithms for contrastive PAC learning, by extending ideas from algorithmic robustness (Diakonikolas & Kane, 2019; Shen & Zhang, 2021; Shen, 2021b; 2023)?

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
