# OpenReview forum: "Towards Efficient Contrastive PAC Learning"
_TMLR — Accepted by TMLR_

### Review · Reviewer_yznr · 2025-03-20

**Summary Of Contributions:**

The author studies contrastive learning of the fundamental concept of linear representations under the PAC learning framework. They are the first one to derive an efficient PAC learning algorithm for contrastive learning.

**Audience:**

Yes

**Claims And Evidence:**

Yes

**Requested Changes:**

How is $\text{EX}(D,h^*)$ defined in Definition 1? Is it a function? If so, does it map from the space $\mathcal{P} \times \mathcal{H}$ of probabilities over $\mathcal{X}$ times the space of models, to the space $\mathcal{U} \times \mathcal{B}$?

There's a typo in eq 1.3, it should be closed curly bracket.

Are $r_F$, $r_1$, and $r_*$ at the bottom of page 3 used to simply distinguish between different radiuses of the considered balls, or do they have another meaning? The first one seems to be the correct answer, but this is only explained later in sections 3.1.1 and 3.1.2.

There's an extra "Then" at the beginning of Lemma 9. More in general, the writing of the paper could be slightly improved.

Some Corollaries are not explicitly proven. While it is easy to see that they are true, I'd suggest to include the proofs in the final version.

**Strengths And Weaknesses:**

The paper is technically sound, and it solves a problem that is -- surprisingly enough -- still open. The writing could be improved. My requested changes are limited, as this is not my primary area of expertise.

---

> ### Author Response · Authors · 2025-03-20
>
> Dear Reviewer yznr,
>
> Many thanks for the prompt and constructive feedback. We will polish the presentation and correct the typos.
>
> Below is a point-to-point clarification on technical questions.
>
> 1. We use EX(D, h^*) to denote the adversary that generates all training and testing samples. Technically, it can be thought of as a function that takes as input sample requests from a learner, and outputs contrastive samples. The way how it returns samples is described in Definition 1.
>
> 2. The reviewer is correct that the subscripts in r (and in other notations) are only used to distinguish hyper-parameters from different types of constraints.
>
> 3. We will add full proofs to corollaries.
>
> Thank you,
>
> Authors

---

### Review · Reviewer_Snpm · 2025-04-19

**Summary Of Contributions:**

The paper considers learning of a Mahalanobis distance function in contrastive learning. The paper introduces the algorithm for efficient learning of the distance, as well as sample complexity for this problem.

**Audience:**

Yes

**Broader Impact Concerns:**

No ethical concerns

**Claims And Evidence:**

No

**Requested Changes:**

In light of the works above, please clarify the novelty of your paper.

**Strengths And Weaknesses:**

As far as I can tell, the theoretical contributions - the algorithm based on convex optimization and sample complexity based on Rademacher complexity - look solid. My main concerns are regarding novelty. I think the authors missed the following papers:
* Regarding sample complexity: ["Sample Complexity of Learning Mahalanobis Distance Metrics" by Verma & Branson](https://www.cs.columbia.edu/~verma/papers/mlearn_nips15.pdf). The paper introduces a similar model (including assumptions on bounded inputs and bounded Frobenius norm of the matrix). Admittedly, they only prove sample complexity for paired comparisons, and then say "the results are easily extended to triplet comparisons." However, their techniques and results look similar.
* Regarding the algorithm: ["Sparse Compositional Metric Learning" by Shi, Bellet, Sha](https://researchers.lille.inria.fr/abellet/papers/aaai14.pdf) (also implemented as a [library](https://contrib.scikit-learn.org/metric-learn/weakly_supervised.html#learning-on-triplets)). Their optimization problem is different, but, as they claim, it's also convex, and it's clearly an ERM. While they don't explicitly consider constraints on the learnable matrix, this can be handled by projected gradient descent.

---

> ### Author Response · Authors · 2025-04-19
>
> Dear Reviewer Snpm,
>
> Thank you for your constructive comments and pointer to two papers. It turns out that the problem setup and main results are incomparable between our work and the two papers.
>
> Our key argument is computationally efficient PAC learning, but Verma and Branson (2015) only presents sample complexity analysis. In fact, even acknowledging that their method can be adapted to contrastive samples, their loss function will end up with coinciding Eq. (3.4) in our submission, for which we have clarified that it is non-convex (thus inefficient). This is exactly where our main techniques, e.g. using SDP, take into play.
>
> The work of Shi, Bellet and Sha (2014) has imposed quite different assumptions. A critical catch is that they assume the distance matrix M can be written as a linear combination of * known * basis matrices. This therefore linearizes the problem. Such technique to achieve convexity is thus entirely distinct from ours - the theoretical guarantees are incomparable as a consequence.
>
> Thank you,
>
> Authors

---

### Review · Reviewer_pE9y · 2025-04-30

**Summary Of Contributions:**

This paper studies the PAC guarantee for contrastive learning under the setting with a fixed $\ell_2$ distance with a linear function class and realizability assumption. It is shown that a complexity of polynomial of 1/\epsion and log 1/\delta is need to guarantee an error less than \epsilon with high probability. This paper is more general than Alon 2024 in that it studies a potentially infinite subspace of the real space, while Alon 2024 focus on finiite \mathcal X; and the approach in Arora 2019 does not apply to obtain the PAC guarantees. On the other hand, this pape focuses on $p=2$ which is less general than the one in Alon 2024.

**Audience:**

Yes

**Broader Impact Concerns:**

None are noted.

**Claims And Evidence:**

Yes

**Requested Changes:**

1. Discuss the extension to large-scale neural networks.
2. Discuss other relaxations to the non-convex problem than the one used in 3.12, and whether they can imply similar generalization results.

**Strengths And Weaknesses:**

Strengths:
1. This paper develops PAC guarantees for the case with linear function class, fixed p=2, which is new in the literature.

Weaknesses:

This paper focuses on the linear function class and assumes realizability. Then the results were established using Rademacher complexity. However, it is commonly known that the approach of  Rademacher complexity is difficult to generalize to large neural networks, which are the ones used in practice. Therefore, a question here is whether the developed methodology can be combined with recent advancements of deep learning theory, e.g., those used in analyzing large-scale neural networks, to obtain guarantees for contrasive learning problems in this paper.

---

> ### Author Response · Authors · 2025-04-30
>
> Dear Reviewer pE9y,
>
> Thank you for the constructive comments. It would be interesting to develop theory for deep learning models, but we cannot promise the possibility of such extension. Such challenge roots in the drastic change in the loss function or equivalently constraints from contrastive samples, i.e. the function g_W(x, y, z) in Eq. (3.1). As we showed, even for linear class, the introduction of contrastive samples has drastically changed the landscape on the computational front.
>
> On the other side, we did discuss extensions to other non-convex problems in Section 4. For example, we expect the problem be efficiently learnable when distance metric is L1-norm. We also believe extension to polynomial threshold functions be possible.
>
> Thank you,
>
> Authors

---

### Author Response · Authors · 2025-05-20
**revision**

Dear Reviewers,

A revision has been uploaded based on our discussion. The change is minor: full proof to a corollary (Corollary 12) is added; typo is corrected; discussion on extending method to neural networks is added (Section 4).

Thank you,

Authors

---

### Comment · Action_Editor_VPHF · 2025-06-08
**More discussion**

Dear authors,

There is one more comment of the reviewer Snpm  regarding Shi, Bellet and Sha (2014):

Comparing with Verma and Branson (2015), they try to argue that their loss function is tractable. Since their final measure is the accuracy, the choice of the loss function is irrelevant for sample complexity. To conclude, their sample complexity bound is not a novel result.

Overall, Mahalanobis distances are the old topic, with a lot of works published in the area, including the ones relying on SDP techniques. The two works I mentioned above are the ones which looked the closest, but, basically, none of the works on Mahalanobis distances are mentioned. Simply put, they don't properly acknowledge related works, which, in my opinion, warrants rejection.

I would like to give you an opportunity to respond to this comment.

AE

---

> ### Author Response · Authors · 2025-06-08
> **response to Reviewer Snpm**
>
> We thank the AE for organizing the discussion and the reviewer's follow-up question.
>
> To Reviewer Snpm:
>
> Q1: Comparing with Verma and Branson (2015), they try to argue that their loss function is [in]tractable. Since their final measure is the accuracy, the choice of the loss function is irrelevant for sample complexity. To conclude, their sample complexity bound is not a novel result.
>
> Re: We cannot follow the argument from the reviewer. Indeed, the argument "Since their final measure is the accuracy, the choice of the loss function is irrelevant for sample complexity" from the reviewer is incorrect: recall that not all loss functions serve as surrogate to error rate, and different loss functions do lead to different sample complexity - to see this, simply recall that Rademacher-based sample complexity depends on the superum of loss function, see e.g.
>
> Theorem 26.12 from Understanding Machine Learning: from Theory to Algorithms, Ben-David and Shalev-Shwartz
>
> Perhaps the reviewer can further clarify this point.
>
> The reviewer seems also not understand the critical difference between computational sample complexity and statistical sample complexity. Without computational concern, PAC learning from contrastive samples is solved by Alon et al 2024 and Verma and Branson (2015) may also imply such result. However, both of these estimators are inefficient, and this work gives the first efficient PAC learner.
>
> Q2: Simply put, they don't properly acknowledge related works, which, in my opinion, warrants rejection.
>
> Re: We added a discussion on Verma and Branson (2015) in the revision - see page 4. The reviewer should, however, understand that linearizing x'W'Wx to x'Gx is such a known technique that we cannot dedicate it to the Mahalanobis distance literature. If the reviewer knows the first work doing so, we are more than happy to cite. As we pointed out in the paper, our contribution lies in properly formulating the contrastive PAC learning problem, applying such SDP technique, and obtaining the computational sample complexity.
>
> After taking a close look at TMLR acceptance criteria (https://jmlr.org/tmlr/acceptance-criteria.html), we feel that our claims are all supported, and we did not over-claim, or took prior work's credits.
>
> Please let us know if you have any follow-up questions.

---

> > ### Comment · Reviewer_Snpm · 2025-06-17
> >
> > > Indeed, the argument "Since their final measure is the accuracy, the choice of the loss function is irrelevant for sample complexity" from the reviewer is incorrect: recall that not all loss functions serve as surrogate to error rate, and different loss functions do lead to different sample complexity.
> >
> > Let's look at Theorem 10 (which you precede with "Now we can prove the main result of this section, the PAC guarantees."). You define the error measure as $err_D(h) = Pr_{(u,b) \sim D} (h(u) \ne b)$, that is, the accuracy, which is indeed the correct measure. In the proof, you use your loss function to bound this error, but the final result of Theorem 10 is the bound on $err_D$. And, as I stated before, such a bound is already known (in fact, if I understood the Verma and Branson correctly, they have a stronger result, since they don't require the large-margin condition).
> >
> > Overall, I would be satisfied if you reframe Section 3.2, to make it clear that your goal is not to prove PAC bounds, but instead to show that you algorithm achieves these complexity bound. As currently written, Section 3.2 looks like it tries to prove novel PAC bounds.
> >
> > > Related work
> >
> > I think you want to cite popular practical approaches (including the one I shared) and their shortcomings.
> >
> > Probably the most important missing reference is *An Efficient Dual Approach to Distance Metric Learning by Chunhua Shen, Junae Kim, Fayao Liu, Lei Wang, Anton van den Hengel*: Formulation in Section 2 seems exactly the same as yours plus regularization.

---

> ### Author Response · Authors · 2025-06-18
>
> Dear Reviewer Snpm,
>
> Regarding the error rate and sample complexity, we believe that the reviewer still did not get the critical difference between statistical sample complexity and computational sample complexity.
>
> Inefficient PAC guarantees are already known, for example, one minimizes 0/1 loss (which is NP-hard), or applies Verma and Branson (which is non-convex, as we clarified in our initial response on April 18).
>
> Efficient PAC guarantees are largely open for contrastive PAC learning, and our main claim -- the first efficient PAC learner -- remains valid.
>
> For more background, inefficient PAC guarantees were known for broad range of problems in robust statistics, but only very recently have efficient PAC guarantees been established. For another example, inefficient agnostic PAC learning is easy while efficient agnostic PAC is known only for certain classes. These highlight the critical difference and phase transition when computational efficiency is taken into account. We refer the reviewer to the following textbook:
>
> Chapter 8, Understanding Machine Learning: from Theory to Algorithms, Ben-David and Shalev-Shwartz
>
> The linearization technique is so far reaching that can even be found in classic textbook on approximation algorithms, though it is still unclear who was the first that invented it. We will add a reference in the revision.

---

> > ### Comment · Reviewer_Snpm · 2025-06-18
> >
> > Again, check Theorem 10, which is your main result. Nowhere in the theorem **statement** you say that you get efficient PAC bounds. You need to rewrite Theorem 10 so that theorem statement implies that you have efficient PAC guarantees.

---

> > > ### Author Response · Authors · 2025-06-18
> > >
> > > Thank you for pointing it out. We have highlighted it in the main statement of Theorem 10, and also added a comment at the end of Section 3.2.

---

### Decision · Action_Editor_VPHF · 2025-06-26

**Recommendation:** Accept as is

**Additional Comments:**

NA

**Audience:**

Yes

**Audience Explanation:**

Contrastive learning is a powerful approach for representation learning, particularly in self-supervised settings, and has also garnered significant interest from the theoretical community. This paper focuses on developing new PAC guarantees for contrastive learning in the supervised setting. It presents several novel theoretical results, which contribute to the understanding of generalization in contrastive learning. As such, the work is likely to attract attention from  theoretical  audiences.

**Claims And Evidence:**

Yes

**Claims Explanation:**

This paper investigates the problem of contrastive PAC learning, with the goal of establishing efficient PAC guarantees. It focuses on a linear representation setting and formulates the task as a binary classification problem—determining whether a pair $(x, y)$ is closer than another pair $(x, z)$. The main contribution lies in analyzing the PAC guarantees of an efficient algorithm based on convex optimization, which involves solving a semidefinite program (SDP) and using eigenvalue decomposition to obtain the linear representation. Generalization bounds are then derived using Rademacher complexity. Reviewers concur that the contributions are solid and significant. Although some noted the limitation to linear settings, the AE considers this work an important theoretical advancement.